# Absolute Position Embedding Learns Sinusoid-like Waves for Attention Based on Relative Position

**Yuji Yamamoto** and **Takuya Matsuzaki**
Tokyo University of Science
1423531@ed.tus.ac.jp    matuzaki@rs.tus.ac.jp

## Abstract

Attention weight is a clue to interpret how a Transformer-based model makes an inference. In some attention heads, the attention focuses on the neighbors of each token. This allows the output vector of each token to depend on the surrounding tokens and contributes to make the inference context-dependent. We analyze the mechanism behind the concentration of attention on nearby tokens. We show that the phenomenon emerges as follows: (1) learned position embedding has sinusoid-like components, (2) such components are transmitted to the query and the key in the self-attention, (3) the attention head shifts the phases of the sinusoid-like components so that the attention concentrates on nearby tokens at specific relative positions. In other words, a certain type of Transformer-based model acquires the sinusoidal positional encoding to some extent on its own through Masked Language Modeling.

## 1 Introduction

The architecture of Transformer (Vaswani et al., 2017) is symmetric with respect to the token position and it captures word order only through the position embedding included in the input. Thanks to this design, Transformer can flexibly learn relationships between tokens while allowing parallelization. To derive a representation of a context, previous language models have used, for instance, recurrent units to process tokens in the given token order, or convolution to aggregate tokens within a given range. In these models, the property of translation invariance of language has been captured through the architecture of the models. In contrast, Transformer receives all tokens at the same time and does not restrict the positions of the tokens on which each token depends; in exchange for it, the translation invariance has to be *learned* rather than imposed by the architecture.

Position embedding is thus the key to make inferences in Transformer context-dependent. While Transformer uses deterministic sinusoidal position embedding, BERT (Devlin et al., 2019) uses a learnable absolute position embedding. The latter learns the positional representation only through word cooccurrences. Clark et al. (2019) and Kovaleva et al. (2019) investigated the attention weights of the self-attention in BERT and found that the attention in some heads is largely determined by relative positions (Figure 1). This implies that even when the *absolute* position embedding is used, self-attention can make an inference depending on *relative* position.

Wang et al. (2021) compared various position embedding methods based on the performance in downstream tasks. Their results showed that local translation invariance and asymmetric position bias with respect to the direction improved the performance. Ravishankar and Søgaard (2021) observed that some columns of absolute position embedding were periodic. Chang et al. (2022) showed that position representation was periodic even in hidden representations. However, it is not clear how the periodicity is used in the model.

In this work, we analyze how attention depends on relative position. As a result, we show that the relative positional dependence of attention emerges due to the following factors.

- The learnable absolute position embedding has sinusoid-like waves with several limited frequencies (§4.1.1).

- Attention heads extract periodic components derived from position embedding from the hidden states. It is made explicit by applying Singular Value Decomposition to the parameters of the pre-trained language model (§4.2).

- The self-attention shifts the phase of the periodic components in the query and the key to decide the direction of attention (§4.3, §4.4).

Thus, it becomes partially clear how the self-attention equipped with learnable absolute position embeddings makes inferences based on context. However, it is suggested that, when the attention is strongly concentrated on the adjacent tokens, the word embedding is also a factor that enables inference based on relative position (§5).

## 2 Background

In this section, we review the multi-head self-attention mechanism and position embedding.

### 2.1 Multi-Head Self-Attention

The self-attention mixes the representations of tokens at different positions. The input to the $l$-th attention layer is a matrix $X_l \in \mathbb{R}^{T \times d}$ whose $k$-th row corresponds to the hidden representation of the $k$-th token as a vector in $\mathbb{R}^d$. The output of the self-attention layer is defined as follows:

$$A_{lh} = \text{softmax}\left( \frac{X_l W_{lh}^Q (X_l W_{lh}^K)^T}{\sqrt{d/n}} \right) \quad (1)$$

$$O_{lh} = A_{lh} X_l W_{lh}^V \quad (2)$$

$$\text{MultiHead}_l = \text{concat}(O_{l1}, \dots, O_{ln}) W_l^O \quad (3)$$

where $W_{lh}^Q$, $W_{lh}^K$, $W_{lh}^V \in \mathbb{R}^{d \times (d/n)}$ and $W_l^O \in \mathbb{R}^{d \times d}$ are the parameters, $n$ is the number of attention heads per layer, and the subscripts $l$ and $h$ are the indices of the layer and the head, respectively.

In this paper, we refer to the $h$-th head in the $l$-th layer as "head($l$, $h$)" for short, and omit the subscripts $l, h$ when the discussions are common to all layers and heads. The matrices $XW^Q$, $XW^K$, and $A$ are called query, key, and attention weight, respectively. If the $(i, j)$ element of $A$ is large, it is interpreted that the $i$-th token in the sentence attends to the $j$-th token.

### 2.2 Position Embedding

Transformer's position embedding is defined as follows (Sinusoidal Position Embedding; SPE):

$$SPE_{(pos, 2i)} = \sin(pos/10000^{2i/d}) \quad (4)$$

$$SPE_{(pos, 2i+1)} = \cos(pos/10000^{2i/d}). \quad (5)$$

Vaswani et al. (2017) hypothesized that it allows the model to easily learn to attend by relative positions, since the offset between two tokens can be expressed as a matrix product:

$$\begin{bmatrix} \sin x_i & \cos x_i \end{bmatrix} \begin{bmatrix} \cos \theta & -\sin \theta \\ \sin \theta & \cos \theta \end{bmatrix}$$
$$= \begin{bmatrix} \sin(x_i + \theta) & \cos(x_i + \theta) \end{bmatrix}. \quad (6)$$

BERT's position embedding is a learnable parameter. It is called the Absolute Position Embedding (APE) because each row represents an absolute position in the input sequence.

RoBERTa (Liu et al., 2019) is identical to BERT in architecture. RoBERTa is pre-trained with 512 tokens for all steps whereas BERT is pre-trained with 128 tokens for 90% of the steps. Hence, RoBERTa's APE recieves the same number of updates regardless of the positions. Wang and Chen (2020) showed that RoBERTa's APE is more orderly than BERT's. We thus use RoBERTa for all experiments to analyze the position dependence of inference.

## 3 Relative Position Dependence of Attention

In this section, we demonstrate that the attention depends on the relative position. Specifically, we analyze in how many heads the attention is dependent on the relative position, and the variation in the direction and concentration of the attention across the heads. The visualizations of the attention weight $A_{lh}$ in Figure 1 shows that, in (a) and (d), each token strongly attends only to the adjacent token, while in (b) and (e), the attention is put on a few nearby tokens on the left or right side. We show that such a pattern is largely invariant with respect to the inputs, by clustering the patterns of the attention on different inputs.

To focus on the relation between the attention and the relative positions, we summarized the attention weight for each input in each head into a vector and applied k-means clustering to them. For that, we defined $t$-offset trace $\text{tr}_t$ as follows:

$$\text{tr}_t(A) = \begin{cases} \sum_{i=1}^{T-t} (A)_{i,i+t} & (t \geq 0) \\ \sum_{i=1}^{T+t} (A)_{i-t,i} & (t < 0) \end{cases} \quad (7)$$

and transformed $A_{lh}$ to a vector:

$$\boldsymbol{a}_{lh} = [\text{tr}_{-10}(A_{lh}), \dots, \text{tr}_{10}(A_{lh})] \in \mathbb{R}^{21}. \quad (8)$$

These vectors represent the trend of attention $A_{lh}$ with respect to relative position.

We input 100 sentences of length 512 created from wikitext-2 (Merity et al., 2017) into RoBERTa and computed vectors $\boldsymbol{a}_{lh}$ for every head. Figure 2 shows the results of applying the k-means to a total of $100 \times 12 \times 12$ vectors $\boldsymbol{a}_{lh}$ when the number of clusters was set to 6. In the clusters named `leftward` and `next-to-left`

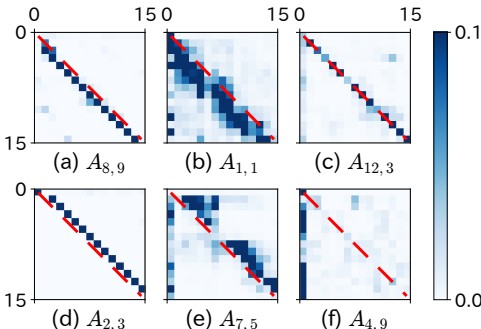

Figure 1: Attention weights for the first 15 tokens of an input text.

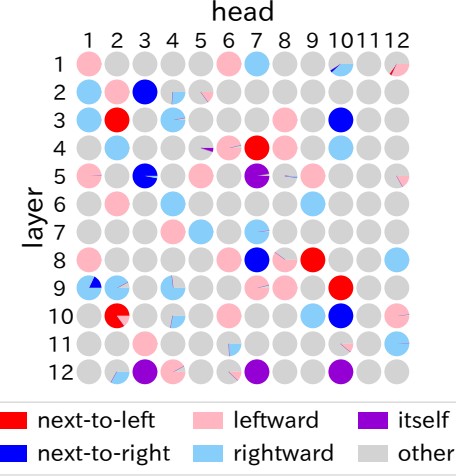

Figure 2: For each head, a pie chart illustrates the percentage of labels assigned to 100 vectors $\boldsymbol{a}_{lh}$ (Eq. (7)).

(resp. `rightward` and `next-to-right`), the attention is concentrated on the nearby tokens on the left (resp. right). Particularly, in the clusters named `next-to-left` and `next-to-right`, the attention is strongly concentrated on the adjacent tokens. We found that, for each head, the vectors $\boldsymbol{a}_{lh}$ corresponding to the 100 inputs were grouped into one or two clusters. This means that there are some heads that always attended to the same direction for all the input.

## 4 Attention to Nearby Tokens

In this section, we show that the attention depends on relative position due to periodic components in the position embeddings. First, we show that the learnable absolute position embeddings acquire several limited frequency components. Second, we show that some attention heads extract the periodicity derived from position embeddings. Finally, we show that the concentration of attention on the nearby tokens is due to the periodicity.

### 4.1 Learned Representation of Positions

First, we show that APE includes sinusoid-like components using the Discrete Fourier Transform (DFT). Next, we show that the position embeddings are confined to a relatively low-dimensional subspace (~15 dimensions) using Principal Component Analysis (PCA). Finally, we show that the encoded positional information occupies a similar number of dimensions in the hidden states as in the position embedding, and the dimensionality becomes smaller in the higher layers using Canonical Correlation Analysis (CCA).

#### 4.1.1 APE Includes Sinusoid-like Waves

We view the RoBERTa's position embedding $E_{pos} \in \mathbb{R}^{T \times d}$ as a collection of $d$ time series of

length $T$. We computed the amplitude spectrum of each column of $E_{pos}$ by applying DFT:

$$\text{spec}_i = \text{abs}(\text{DFT}(E_{pos}\, \boldsymbol{e}_i)) \qquad (9)$$

where $\boldsymbol{e}_i \in \mathbb{R}^d$ is the $i$-th elementary unit vector. The vector $E_{pos}\boldsymbol{e}_i$ is hence the $i$-th column of $E_{pos}$.

Figure 3a shows the mean and the quartile range of the amplitude spectra of $d = 768$ time series. The amplitude spectra peak at some frequencies, indicating that the learnable APE has acquired periodicity through pre-training, even though it is not explicitly defined using sinusoidal waves as in Transformer. In contrast, there are no peaks in the amplitude spectrum of a word embedding sequence of a sample text.[1] The periodicity is thus an unique property of learned position embeddings.

We investigated whether similar properties are present in pre-trained models other than RoBERTa (Figure 3b and 13). The amplitude spectra of the encoder-only models are similar to RoBERTa regardless of language, but ones of GPT-2 (Radford et al., 2019) are higher at lower frequencies.

The decoder model probably can focus attention on the neighboring tokens of itself without periodicity-based mechanisms (§4.3). For example, if attention is paid more strongly to the backward token, in BERT, attention is focused on the end of the sentence, but in a decoder with causal attention mask, attention is focused on itself. We leave it to future research to clarify whether this

---

[1]The text taken from abstract and introduction of (Vaswani et al., 2017) was used as input to create the figures in this paper, unless otherwise stated.

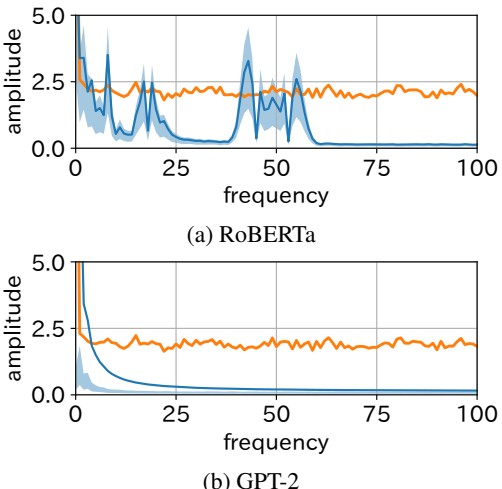

(a) RoBERTa

(b) GPT-2

Figure 3: Amplitude spectra of position embeddings. Blue line is the mean of $\mathrm{spec}_i$ and blue area is the quartile range. Orange line is the spectrum of word embeddings for an input before adding position embeddings.

phenomenon really occurs in the causal model, and in this paper we focus on the case where self-attention is symmetric with respect to position.

### 4.1.2 Dimensionality of Positional Representation

We applied PCA to the position embeddings to see how many dimensions were used to represent the positions. The cumulative principal component contribution rate of the position embedding was 50.51% for the top 4 dimensions, and 92.23% for the top 12 dimensions (see Figure 12 in Appendix A for a plot). This means that the positions are mostly encoded in a low dimensional subspace.

We then employed CCA to show how much the input to the self-attention layers included the positional representation. CCA is a method for investigating the relation of two sets of variables by finding the parameters $\boldsymbol{a}$ and $\boldsymbol{b}$ that maximizes the correlation between two synthetic variables $X\boldsymbol{a}$ and $Y\boldsymbol{b}$ given two inputs $X$ and $Y$. Raghu et al. (2017) showed that CCA allows layer-to-layer comparison of deep learning models. We used the representation of neurons and layers proposed by them and computed the correlation between the hidden layer and the position embeddings.

In this study, the $i$-th neuron $\boldsymbol{z}_i^l$ of layer $l$ and the $l$-th layer $Z^l$ are represented as follows:

$$\boldsymbol{z}_i^l = \left[ z_i^l(\boldsymbol{x}_1), \ldots, z_i^l(\boldsymbol{x}_m) \right]^T \qquad (10)$$

$$Z^l = \left[ \boldsymbol{z}_1^l, \ldots, \boldsymbol{z}_n^l \right] \qquad (11)$$

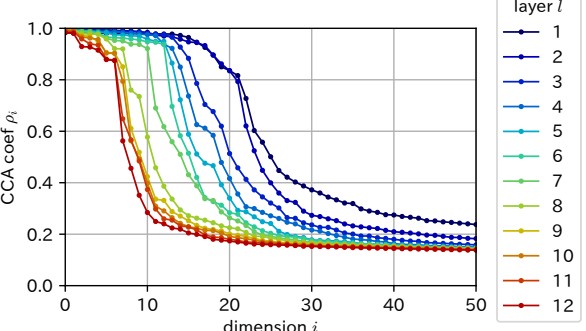

Figure 4: CCA coefficient for each layer compared to position embedding.

where $z_i^l(\boldsymbol{x}_j)$ is the response of the $i$-th neuron to input $\boldsymbol{x}_j$. We input 200 sentences of length 512 created from wikitext-2 into RoBERTa and collected the responses of each layer for the input of $m = 200 \times 512$ tokens. We then maximized their correlation coefficients $\rho_k$:

$$\rho_k = \max_{\boldsymbol{a}_k, \boldsymbol{b}_k} \mathrm{Corr}(Z^{l_1}\boldsymbol{a}_k, Z^{l_2}\boldsymbol{b}_k) \qquad (12)$$

where $\boldsymbol{a}_k$ is chosen such that it is orthogonal to $\boldsymbol{a}_1, \ldots, \boldsymbol{a}_{k-1}$ and similarly for $\boldsymbol{b}_k$. The CCA coefficients in Figure 4 show that the hidden states of higher layers have lower correlations with the position embeddings. This is in agreement with Lin et al.'s (2019) result that BERT phases out positional information as it passes through the layers.

The CCA result indicates that the number of components that are highly correlated with position embedding is only 5~20-dimensions, and the PCA result suggests that it is enough to accommodate the positional representation. This indicates that the hidden states include positional representation in a low-dimensional subspace, similarly to position embedding.

### 4.2 Positional Representation in Self-Attention

It is not at all clear how the positional representation in the hidden states contribute to the inference in self-attention. We thus analyzed how attention weight is calculated, which is one of the most important process in self-attention.

### 4.2.1 Rethinking About Query and Key

The attention weight $A$ is determined by the inner product of the rows of the query and the key matrix. We thus expect that the relative position dependence of the attention can be explained by analyzing the relationship between the query and

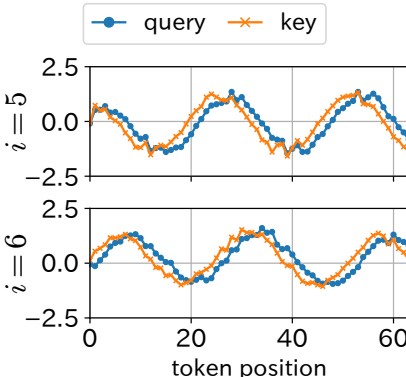

Figure 5: Column $i$ of the query and the key in head(1, 1).

the key. We begin by rethinking about their definition. The two parameter matrices $W^Q$ and $W^K$ contribute to the attention only through their product $W^Q(W^K)^T$ (Eq. (1)). Hence, they can be seen as a parametrization of a rank-restricted $d \times d$ matrix, and we may consider another decomposition of the product $W^Q(W^K)^T$.

We found that some sinusoid-like components were obtained from the query and the key by applying Singular Value Decomposition (SVD) to $W^Q(W^K)^T$. Specifically, the query and the key are redefined as the product of the hidden state and the singular vectors:

$$W^Q(W^K)^T = U^Q S (U^K)^T \tag{13}$$

$$Q = X U^Q, \quad K = X U^K \tag{14}$$

where the matrix $S \in \mathbb{R}^{d \times d}$ is a diagonal matrix $\text{diag}(s_1, \ldots, s_{d/n}, 0, \ldots, 0)$ and each element $s_i$ is a singular value. In the rest of this paper, we refer to $Q$ and $K$ defined above as the query and the key, respectively. As shown in Figure 5, in the head in which the attention depends on relative position, sinusoid-like waves appear in several columns of the redefined query $Q$ and the key $K$. Moreover, a sine wave is paired with a cosine wave of the same frequency, as the 5-th and the 6-th column for head(1, 1) shown in Figure 5.

Furthermore, the introduction of SVD provides a new theoretical interpretation of self-attention. Let $R$ be the orthogonal matrix $R = (U^Q)^T U^K$. Then $U^K$ can be written as $U^K = U^Q R$ due to the orthogonality of the singular vectors. Thus, the key $K$ can be written as:

$$K = X U^K = X U^Q R = QR \tag{15}$$

That is to say, the rows of the key are the result of an orthogonal transformation of the rows of the

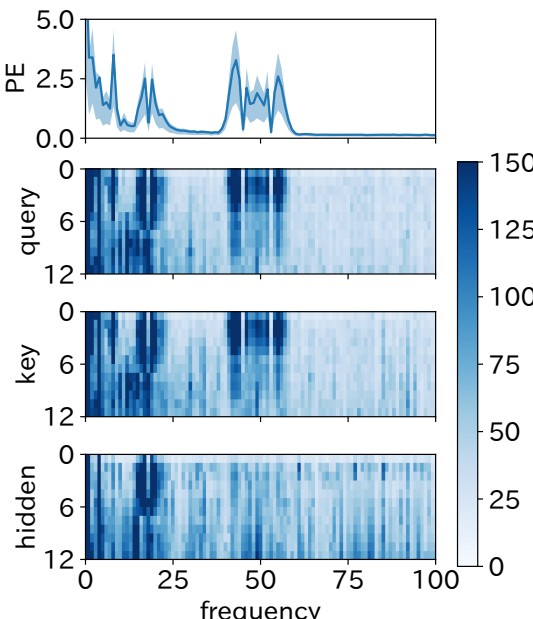

Figure 6: The frequency-wise maxima of the amplitude spectra. The $l$-th row of the heat map corresponds to max-spec$_l$. The top figure is a reiteration of Figure 3a.

query. The relation between the query and the key is thus summarized in the matrix $R$.

In addition, since the singular value matrix $S$ is diagonal, the product of the query and the key can be written as follows:

$$XW^Q(XW^K)^T = QSK^T = \sum_{i=1}^{d/n} s_i \boldsymbol{q}_i \boldsymbol{k}_i^T \tag{16}$$

where $\boldsymbol{q}_i$ and $\boldsymbol{k}_i$ are the $i$-th columns of $Q$ and $K$, respectively. Eq. (16) says that the subsets of queries $[\boldsymbol{q}_i]$ and keys $[\boldsymbol{k}_i]$ corresponding to the top singular values $[s_i]$ are more important in the calculation of the attention distribution. We hereafter call the matrix $QSK^T$ *attention score* (i.e., attention before applying softmax).

### 4.2.2 Spectral Analysis of Query and Key

We computed the amplitude spectrum of each column of the query $Q_{lh}$ in each head, using the same procedure for the position embeddings in §4.1.1:

$$\text{spec}_{lhi} = \text{abs}(\text{DFT}(Q_{lh}\boldsymbol{e}_i)) \tag{17}$$

$$\text{max-spec}_l = [\max_{h,i}(\text{spec}_{lhi})_f]_{f=0,\ldots,T/2}. \tag{18}$$

By taking the maximum of $\text{spec}_{lhi}$ among the heads and the columns of $Q_{lh}$, we check if there are high peaks in the spectra of the queries in the $l$-th layer (similarly for the keys and hidden states).

Figure 6 show that the query and key spectra peak at the same frequency bands as the position embeddings, indicating that attention heads extract periodic representations derived from the position embeddings. Furthermore, some of the peaks disappear in the higher layers (downward in the Figure 6), indicating that the periodic components derived from the position embeddings are not dominant in the higher layers. This is consistent with the result that the correlation between the position embeddings and the hiddem states of each layer gradually decreases (§4.1.2). It also agrees with the result shown by Lin et al. (2019) that position information is discarded between the 3rd and 4th layers.

### 4.3 Attention Based on Relative Position is due to the Phase Shift

As shown in Figure 5, there are phase shifts in the sinusoid-like components of the query and the key. In this subsection, we focus on the phase shifts, and clarify that the direction and the width of the phase shifts determine where the attention is concentrated. We measure the direction and the width of the phase shift by cross-covariance and cross-correlation, defined as follows:

$$\text{xcov}_j(t) = \begin{cases} \sum_{i=1}^{T-t} q_{i,j}k_{i+t,j} & (t \geq 0) \\ \sum_{i=1}^{T+t} q_{i-t,j}k_{i,j} & (t < 0) \end{cases} \quad (19)$$

$$\text{xcorr}_j(t) = \frac{\text{xcov}_j(t) - \mathbb{E}_t[\text{xcov}_j(t)]}{\|\boldsymbol{q}_j\|\|\boldsymbol{k}_j\|} \quad (20)$$

For example, Figure 7 shows the cross-correlation between the queries and keys in Figure 5. Both $\text{xcorr}_5(t)$ and $\text{xcorr}_6(t)$ attain a maximal value at $t = -2$ and $t = -3$. It means that the phase shift between the queries and the keys are approximately 2.5 tokens.

It can be explained theoretically that the direction of phase shift coincides with the direction of attention. The cross-covariance is related to the product of query and key (Eq. (16)):

$$\text{tr}_t(QSK^T) = \sum_{i=1}^{T-t}(QSK^T)_{i,i+t} \quad (21)$$

$$= \sum_{i=1}^{T-t}\sum_{j=1}^{d/n} s_j q_{i,j}k_{i+t,j} \quad (22)$$

$$= \sum_{j=1}^{d/n} s_j \text{xcov}_j(t) \quad (23)$$

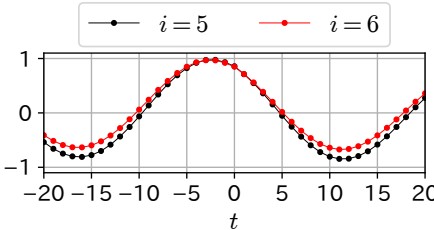

Figure 7: The cross-correlation $\text{xcorr}_i(t)$ between the queries and the keys in Figure 5.

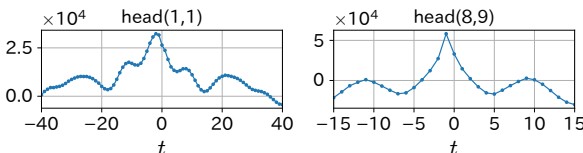

Figure 8: The sum of cross-covariances weighted by singular values (Eq. (23))

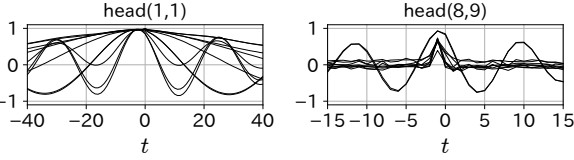

Figure 9: The cross-correlations $\text{xcorr}_j(t)$ corresponding to the top 10 singular values.

where $\text{tr}_t$ is the $t$-offset trace defined in Eq. (7) and $s_1, \ldots, s_{d/n}$ are the singular values of $W^Q(W^K)^T$. According to Eq. (21-23), the sum of the attention scores at relative position $t$ is equal to the weighted sum of the inner product of $\boldsymbol{q}_j$ and $\boldsymbol{k}_j$ shifted by $t$. Therefore if the $\text{xcov}_j(t)$s corresponding to the top singular values attain maximal values at the same $t$, the attention is likely to be concentrated around relative position $t$.

Figure 8 and 9 show that it is actually the case. Figure 9 presents the cross-correlations of the query $\boldsymbol{q}_j$ and the key $\boldsymbol{k}_j$ for $j$s corresponding to the top singular values. We can see how the concentration of attention on nearby tokens is formed by the weighted superposition of the cross-correlations. In head(1, 1), the maxima of the cross-correlations near $t = -1$ stack up, while the maxima away from $t = -1$ cancel with the minima of other components. Since there are multiple periodic components with different frequencies (Figure 3a), the attention is not concentrated away from each token. In contrast, in head(8, 9), some cross-correlations have narrow peaks only at $t = -1$, and it makes the attention be very concentrated only on the adjacent tokens. However, it is unlikely that the cause of the narrow peaks

is the sinusoid-like waves, because their period is approximately 8 tokens or more.[2]

## 4.4 Phase Shift Width is the Same even if Frequencies are Different

We saw in the previous subsection that the main factor of the concentration of attention is that the phases of multiple sinusoid-like waves are all shifted by the same number of tokens. In this subsection, we explain it based on the property of the eigenvalues and the eigenvectors of the matrix $R$ that relates the query $Q$ to the key $K$.

Let $p_i \in \mathbb{C}^d$ and $\lambda_i \in \mathbb{C}$ be an eigenvector and an eigenvalue of the matrix $R$, respectively. Since $R$ is an orthogonal matrix, its eigenvalue can be expressed as $\lambda_i = e^{j\theta_i}$ where $j$ is the imaginary unit. From Eq. (15), the following relation between the query and the key holds:

$$K p_i = QR p_i = Q\lambda_i p_i = Q p_i \cdot e^{j\theta_i}. \quad (24)$$

Let two conditions be assumed: (1) for each $i = 1, \ldots, d$, the time series $Q p_i$ is sinusoidal with a single frequency $f_i$ and (2) the ratio of the argument $\theta_i$ of the eigenvalue $\lambda_i$ to the frequency $f_i$ is constant regardless of $i$. Then Eq. (24) implies that the phases of $Q p_i$ and $K p_i$ differ by a constant number of tokens for any eigenvector $p_i$:

$$(Q p_i)_t = (K p_i)_{t+\Delta} \quad (25)$$

where $\Delta = (T/2\pi) \cdot (\theta_i/f_i)$. Furthermore, $Q_t = K_{t+\Delta}$ follows from Eq. (25). We provide the proofs in Appendix D.

We verify whether the two conditions hold by analyzing the periodicity of $Q p_i$ and the ratio of the frequency to the argument $\theta_i$ of the eigenvalue $\lambda_i$. To do so, we define the bivariate functions $g$ for frequency and argument as follows:

$$g(f, \theta_i) = \mathrm{abs}(\mathrm{DFT}(Q p_i))_f \quad (26)$$

This function $g$ is shown in Figure 10 as a 2D histogram. Figure 10 shows that the spectrum of $Q p_i$ has peaks in different frequency bands according to $\theta_i$. It means that the component along each eigenvector $p_i$ is fairly band-limited, namely they are sinusoid-like. Furthermore, the spectral peaks

[2]Figure 3a shows that the maximum frequency of the position embeddings is around 60. The minimum period is hence $512/60$ ($> 8$) tokens.

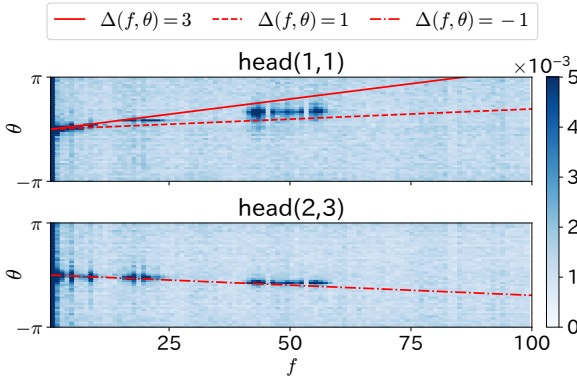

Figure 10: 2D histogram of the function $g(f, \theta)$. The section at $\theta = \theta_i$ is the amplitude spectrum of $Q p_i$.

appear around a straight line. Specifically, the ratio $\Delta(f, \theta)$ of frequency to phase defined as follows is nearly constant:

$$\Delta(f, \theta) = \frac{T}{2\pi} \cdot \frac{\theta}{f}. \quad (27)$$

Hence, the two conditions are in fact approximately hold. Thus, by multiplying the matrix $R$, the sinusoid-like components in the key are offset by a certain number of tokens relative to the query.

## 5 Attention to the Adjacent Tokens

Figure 9 shows that, in head(8, 9), there are not only sinusoid-like waves but also spikes at the relative position $t = -1$. In this section, we dive deeper into the fact that, in some heads, the attention is focused only on the adjacent token. We analyze how the attention changes when we modify either the position embedding or the word embedding component. The following are the results on the modified inputs and insights from them.

**Baseline** The input is the sum of word embedding and position embedding as usual. This result is shown in Figure 8 and 9.

**Only position embedding** The word embeddings are replaced with zero vectors. Figure 11a shows that spikes of cross-correlation do not appear for this case. This suggests that position embedding is not the only cause of the spikes.

**Only word embedding** The position embeddings are replaced with zero vectors. Figure 11b shows that most of the cross-correlations are flat for this case. It suggests that word embedding contributes less to make the attention dependent on relative position. However,

it is interesting that the cross-covariance at $t = -1$ (i.e., left neighbor) is relatively large even without position information.

**Shuffling word embeddings** The order of word embeddings is shuffled. Figure 11c shows that spikes appear at the relative position $t = -1$ even for this case. It suggests that the contribution of position embedding is significant in determining the direction of attention, since the attention is focused on the preceding word regardless of what it actually is.

As mentioned at the end of Section 4.3 and also suggested by Figure 11a, it is unlikely that position embedding is the only cause of the strong concentration of attention on the adjacent tokens. However, if the word embeddings identified the adjacent tokens (i.e., if the narrow peak of cross-correlation appeared due to an interaction of the word embeddings of a frequently occurring bigram), the attention would have been moved to non-neighboring positions by the shuffling of word embeddings, but this was not the case. It is thus suggested that the concept of adjacency in RoBERTa is built upon both word meaning and positional information.

# 6 Remark on the Learning Process of Position Embeddings

The training data of the masked language modeling is only word sequences without positional information. It suggests that the relative position dependence of attention is acquired by the combination of two factors: (1) the linguistic property that related words tend to appear nearby due to grammatical rules and collocations, and (2) the property that attention is focused on the word that is syntactically or semantically related and hence gives clue to fill the masked token. In appendix E, we demonstrate that the position representation can be acquired with a much smaller amount of training data than pre-training.

# 7 Related Works

Ravishankar and Søgaard (2021) observed that some columns of absolute position embedding were periodic and sinusoidal position embedding was more effective in multilingual models than other embedding methods. Chang et al. (2022) showed that position representation was periodic even in hidden representations by using Linear

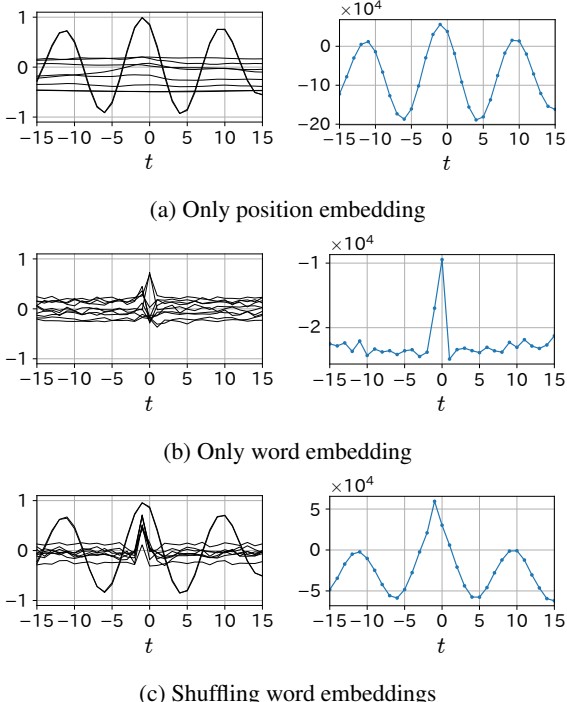

(a) Only position embedding

(b) Only word embedding

(c) Shuffling word embeddings

Figure 11: The cross-correlations (left) and the weighted sum of cross-covariances (right) in head(8, 9) when we modify either the position embedding or the word embedding component of the input.

Discriminant Analysis, i.e., by identifying the axes separating the different position representations. We showed that sinusoid-like components could be extracted from the hidden states by applying SVD to the pre-trained parameters of the self-attention even though SVD does not have the objective of separating positional representations.

Vaswani et al. (2017) stated that the sinusoidal position embedding allowed position offsets to be represented by a rotation transformation (Eq. (6)), and this could prompt learning of attention that depends on relative position. We showed that the query and the key in the self-attention included sinusoid-like components and the learned parameters in the self-attention shift the phase of the query and the key relatively. This means that the mechanism hypothesized by Vaswani et al. is in fact acquired through pre-training by masked language modeling. However, the frequencies acquired by the position embedding of RoBERTa are only in the specific frequency bands, whereas sinusoidal position embedding has $d/2$ frequencies (Eq. (4)-(5)). RoBERTa thus seems to have acquired a more economical positon embedding than sinusoidal position embedding.

Recently, there are many variants of position

embedding and Dufter et al. (2022) surveyed them exhaustively. In particular, Rotary Position Embedding (RoPE) (Su et al., 2022), a type of Relative Position Embedding (RPE) (Shaw et al., 2018), relates to the property that the self attetnion acquires the rotation matrix through pre-training. To acquire relative positional dependence, RoPE widens the angle between query and key proportionally to the relative position, while pre-trained self-attention rotates the hidden states containing absolute positional bias by the same angle regardless of position. In other words, APE and self-attention, which are distant components, must acquire frequency and angle of rotation, respectively, to satisfy the relation Eq. (27). If translation invariance is essential to language understanding, the cost of discovering this relation is a possible reason why APEs are inefficient to learn compared to RPEs.

## 8 Conclusion

We analyzed the concentration of attention based on relative position in the self-attention using the learnable absolute position embedding. As a result, we showed that it is due to relative phase shifts of the periodic components in the query and the key derived from position embeddings. Our results explain in part that absolute position embedding allows inference based on relative position.

## 9 Limitations

As mentioned in §4.1.1, the positional representation of the GPT-2 differs significantly from ones of the encoder models. Thus, it is impossible to interpret the inferences of the decoder-only model in the same way as in this study.

In the tasks of predicting sentence structure (e.g., dependency parsing), the relative order of two tokens is important regardless of the distance between them. However, we analyzed the dependence of the output of each token only on the nearby tokens. Thus, it is unclear whether position embedding provides relative position information that helps determine which of the distant tokens precedes the other.

We obtained an interpretable representation by decomposing the attention scores before applying softmax function (Eq. (16)). When analyzing the contribution of the decomposed representations to downstream components (e.g., Value in self-attention), the non-linearity of softmax function should be taken into account.

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

## A Cumulative Principal Component Contribution Rate

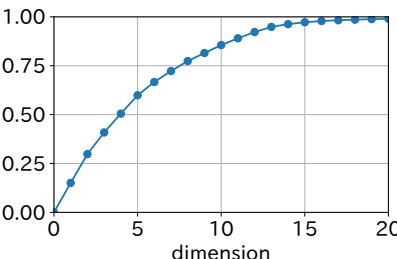

Figure 12: Cumulative principal component contribution rate of RoBERTa position embedding.

## B Amplitude Spectra of Various Models

We computed the amplitude spectra of the position embeddings of `bert-base-uncased`, `cl-tohoku/bert-base-japanese-whole-word-masking`, `xlm-roberta-base`, and `gpt2` in the same way as Figure 3a. In Figure 13, the encoder models BERT and RoBERTa both have peaks in their amplitude spectra, and RoBERTa has a higher peaks. On the other hand, the GPT-2 decoder model has only low-frequency components. This suggests that the representation of absolute position embedding is similar between encoder models, regardless of language, but differs significantly between encoder models and decoder models. In fact, Irie et al. (2019) and Kazemnejad et al. (2023) showed that the position embeddings is unnecessary for the decoder models.

## C Comparing Different Architectures

This paper analyzed an encoder-only model (RoBERTa). In this section, we apply our methods to other transformer-based architectures: decoder-only and encoder-decoder. The target pre-trained models are GPT-2 (gpt2) for decoder-only and BART (facebook/bart-base) (Lewis et al., 2020) for encoder-decoder, both of which use absolute position embedding and are available on hugging-face. To reduce the gap between architectures, when analyzing the decoders, we use the attention scores, which is the matrix before causal masking and softmax function are applied, instead of the attention weights.

The attention matrices of GPT-2 show two main patterns: one related to position and the other not. Figure 14a shows that the attention head of GPT-2 pays stronger attention to the closer tokens by

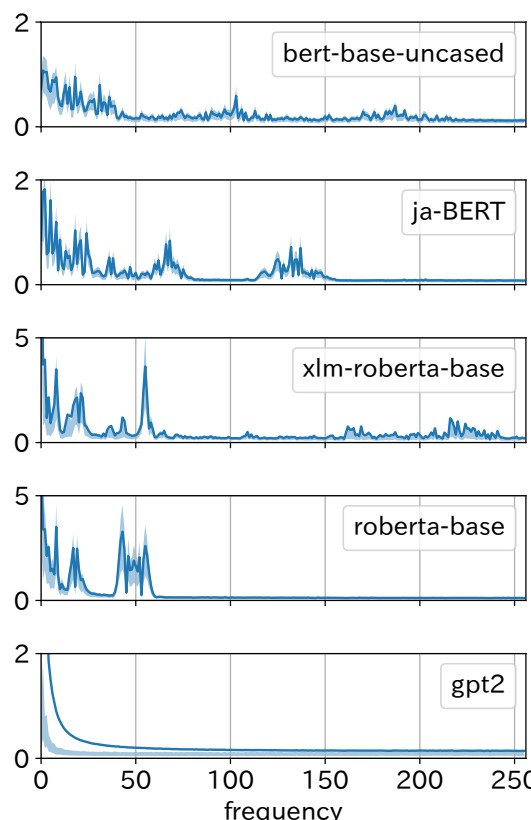

Figure 13: Amplitude spectra of position embedding of various models.

concentrating attention on the backward tokens, which are masked in the subsequent processing. Figure 15 shows the result of clustering the attention scores of GPT-2 using k-means. The number of clusters was set to 2 and the inputs to k-means were $t$-offset traces (Eq.(8)) from $t = -30$ to 30. The heads that depend on position are found in the lower layers.

Along §4.4, we investigate the relationship between the frequency of the hidden state and the angle of the rotation matrix inherent in the parameters of attention head. Figure 16a differs from the case of RoBERTa (Figure 10) in that the peaks appear horizontally rather than holding a constant ratio between frequency and angle. Furthermore, according to Figure 13, the dominant components in position embedding of GPT-2 are those with frequencies below 10, i.e., with periods longer than $51.2 (= 512/10)$ tokens. These differences from RoBERTa imply that even when the attention is focused within a few tokens in a particular head, it cannot be attributed to the mechanism described in §4.3.

In BART, the tendency in the positional dependence of attention differs between encoder and

decoder. In encoder, the patterns of attention weights (Figure 17) and their trend (Figure 18), such as direction and strength, are similar to those of RoBERTa (Figure 1 and 2). On the other hand, comparing Figure 14 and Figure 20, the patterns of decoder is different from those of GPT-2. Such patterns in the BART decoder could not be clustered based on the strength or direction of attention using k-means. Like in GPT-2, Figure 21 shows that the peaks appear horizontally in the decoder.

## Figures for GPT-2

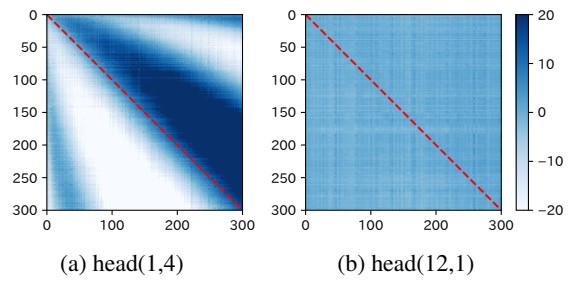

(a) head(1,4)  (b) head(12,1)

Figure 14: Attention scores of GPT-2 for the first 300 tokens of an input text.

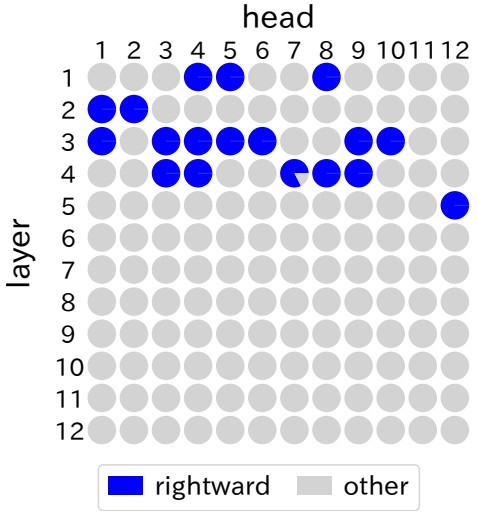

Figure 15: Trend of attention scores in GPT-2.

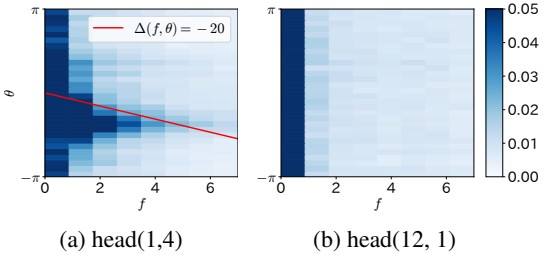

(a) head(1,4)  (b) head(12, 1)

Figure 16: 2D histogram of the function $g(f,\theta)$ in GPT-2.

## Figures for BART encoder

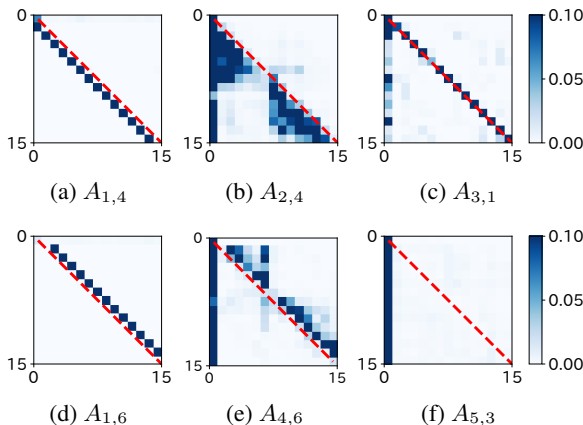

(a) $A_{1,4}$  (b) $A_{2,4}$  (c) $A_{3,1}$

(d) $A_{1,6}$  (e) $A_{4,6}$  (f) $A_{5,3}$

Figure 17: Attention weights of BART encoder for the first 15 tokens of an input text.

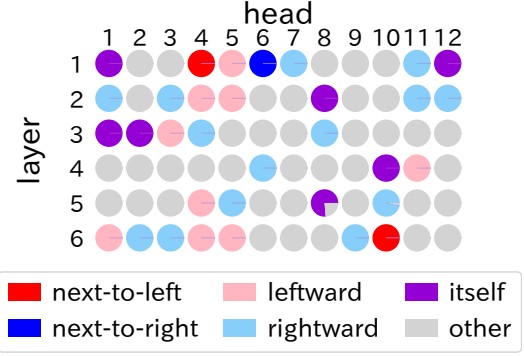

| | next-to-left | | leftward | | itself |
| --- | --- | --- | --- | --- | --- |
| | next-to-right | | rightward | | other |

Figure 18: Trend of attention weights in BART encoder.

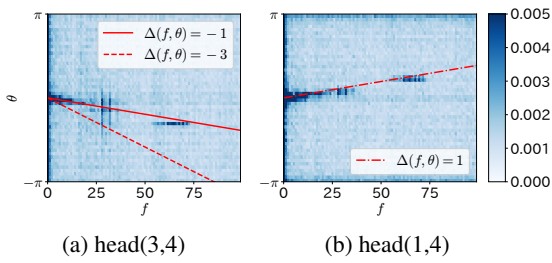

$\Delta(f,\theta) = -1$
$\Delta(f,\theta) = -3$
$\Delta(f,\theta) = 1$

(a) head(3,4)  (b) head(1,4)

Figure 19: 2D histogram of the function $g(f,\theta)$ in BART encoder.

## Figures for BART decoder

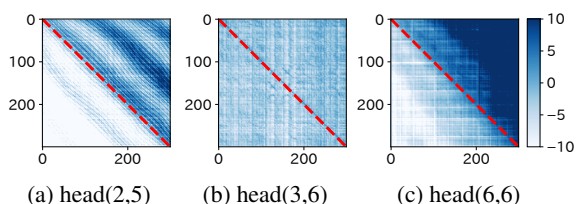

(a) head(2,5)  (b) head(3,6)  (c) head(6,6)

Figure 20: Attention scores of BART decoder for the first 300 tokens of an input text.

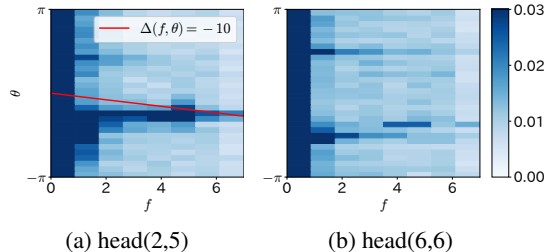

(a) head(2,5)   (b) head(6,6)

Figure 21: 2D histogram of the function $g(f,\theta)$ in BART decoder.

## D Theorems About the Phase Shift Between the Query and the Key

**Theorem 1.** *The following two assumptions are given:*

1. *$Q\boldsymbol{p}_i$ is sinusoidal with a single frequency $f_i$.*

2. *The ratio of the argument $\theta_i$ of an eigenvalue $\lambda_i$ to the frequency $f_i$ is constant regardless of $i$.*

*In this case, Eq. (25) is derived from Eq. (24):*

$$K\boldsymbol{p}_i = Q\boldsymbol{p}_i \cdot e^{j\theta_i} \qquad (24)$$
$$\implies (Q\boldsymbol{p}_i)_t = (K\boldsymbol{p}_i)_{t+\Delta} \qquad (25)$$

*Proof.* Since $Q\boldsymbol{p}_i$ is sinusoidal, the $t$-th element in polar form is as follows:

$$(Q\boldsymbol{p}_i)_t = r_i \exp[j2\pi f_i t/T] \qquad (28)$$

where $r_i$ is the absolute value of $(Q\boldsymbol{p}_i)_t$. Then the $t$-th element of $K\boldsymbol{p}_i$ can be written as follows from Eq. (24):

$$(K\boldsymbol{p}_i)_t = r_i \exp[j2\pi f_i t/T] \cdot \exp[j\theta_i] \qquad (29)$$
$$= r_i \exp[j(2\pi f_i t/T + \theta_i)] \qquad (30)$$
$$= r_i \exp[j2\pi f_i(t + \Delta_i)/T] \qquad (31)$$

where $\Delta_i = (T/2\pi) \cdot (\theta_i/f_i)$. Now, since the ratio of the argument $\theta_i$ to the frequency $f_i$ is constant, $\Delta_i$ is independent of $i$, namely, $\Delta_i$ can be replaced with a constant $\Delta$. Thus, $(Q\boldsymbol{p}_i)_t = (K\boldsymbol{p}_i)_{t+\Delta}$ holds from Eq. (28) and Eq. (31). $\square$

Let the matrix $P = [\boldsymbol{p}_1, \boldsymbol{p}_2, \ldots, \boldsymbol{p}_d]$, then the following equation holds:

$$(QP)_t = (KP)_{t+\Delta}. \qquad (32)$$

**Theorem 2.** *When, in a given basis, the position of each token in the key is offset by $\Delta(\in \mathbb{Z})$ tokens relative to the query, then in any basis, there is an*

*offset of $\Delta$ tokens between the query and the key. That is,*

$$\exists P \in \mathbb{C}^{d\times d} \ s.t. \ |P| \neq 0 \ and \ (QP)_t = (KP)_{t+\Delta} \qquad (33)$$
$$\implies \forall B \in \mathbb{C}^{d\times n}, \ (QB)_t = (KB)_{t+\Delta} \qquad (34)$$

*where $Q, K \in \mathbb{R}^{T\times d}$ are the query and the key respectively and $n \in \mathbb{N}$ is less than or equal to $d$. The subscript $t$ indicates the $t$-th row of the matrix.*

*Proof.* Let $B$ be an arbitrary matrix. Then,

$$(QB)_{ti} = (QPP^{-1}B)_{ti} \qquad (35)$$
$$= \sum_{k=1}^{d}(QP)_{tk}(P^{-1}B)_{ki} \qquad (36)$$
$$= \sum_{k=1}^{d}(KP)_{t+\Delta,k}(P^{-1}B)_{ki} \ (\because \text{Eq. (33)}) \qquad (37)$$
$$= (KPP^{-1}B)_{t+\Delta,i} \qquad (38)$$
$$= (KB)_{t+\Delta,i} \qquad (39)$$

$\square$

In particular, when $B = I$, the equation $Q_t = K_{t+\Delta}$ is obtained.

## E How the Relative Position Dependence of Attention Emerges

How does a masked language model acquire the concept of "neighborhood" even though absolute position embedding learns without the information about the order of the tokens? In this section, we demonstrate the process of acquiring the ability to focus attention on nearby tokens by re-learning the position embedding under the masked language modeling.

Suppose that the following sentence is input into a pre-trained model:

"This <unk> is <unk> an <unk> <mask> ."

Since this sentence is collapsed with many UNK tokens, the model cannot fill the MASK correctly. If we re-learn the position embedding, will the model be able to fill it?

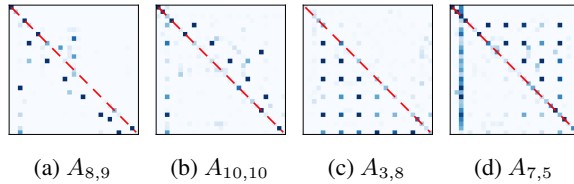

|  | (a) $A_{8,9}$ | (b) $A_{10,10}$ | (c) $A_{3,8}$ | (d) $A_{7,5}$ |

Figure 22: Attention weight with relearned position embedding

| PE | predicted top-5 tokens |
|---|---|
| relearned | exercise essay experiment article adventure |
| RoBERTa base | and the , . to |

Table 1: Top 5 MASK candidates when 3 UNKs are put between each token in "This is an MASK."

## E.1 Experiment

We additionally train RoBERTa with the following configurations.

**Dataset** We used 8645 samples consisting of more than 128 tokens from wikitext-2.

**Putting UNK token** Three UNK tokens were inserted after each token in all sentences; the token sequence $[t_1, t_2, ..., t_{128}]$ was expanded to $[t_1$, unk, unk, unk, $t_2$, unk, unk, unk, ..., $t_{128}$, unk, unk, unk].[3]

**Freezing parameters** We freezed all parameters except position embeddings. This encourages the position embeddings to learn to focus attention only on position $t = 1, 5, 9, 13, \cdots$ and prevents position-independent shortcut solutions (e.g., shortening the UNK embedding to ignore it).

**Position embedding** The position embeddings were initialized with random values following a normal distribution $\mathcal{N}(0, 0.02^2)$ before training.

## E.2 Results

Table 1 shows that, before the additional training, RoBERTa cannot fill MASK with the appropriate tokens (in this case, nouns beginning with a vowel) when UNKs are inserted, but after the additional

training, the models can predict them. The attention weights during inference are visualized in Figure 22. Since the attentions are concentrated every fourth token, the model with relearned position embeddings recognizes the relative positions of the non-UNK tokens. This result suggests that the attention is not focused on the tokens that co-occur frequently, but on the tokens that are informative to fill the MASK token. Thus, the relative position dependence of attention is simply caused by the fact that related words appear nearby.

In summary, this experiment indicates that the relative position dependence of attention is caused by the combination of two factors: (1) the linguistic property that related words tend to appear nearby due to grammatical rules and collocations, and (2) the property that attention is focused on words that are related in some sense.

---

[3] Since Transformer is symmetric with respect to position, this process is similar to padding. However, the PAD token of RoBERTa published on hugging face is zero-vector, which is inconvenient for analyzing interactions between word embeddings. Thus, we choose UNK tokens.