# OpenReview forum: "Absolute Position Embedding Learns Sinusoid-like Waves for Attention Based on Relative Position"
_EMNLP/2023/Conference — EMNLP 2023 Main_

### Official Review · Reviewer_8B3x · 2023-08-02

**Soundness:** 5

**Excitement:**

4: Strong: This paper deepens the understanding of some phenomenon or lowers the barriers to an existing research direction.

**Paper Topic And Main Contributions:**

This paper mainly discusses the behavior of attention mechanisms in pre-trained language models, specifically focusing on the impact of position embeddings on attention. The authors conduct a series of experiments to analyze the relationship between attention and position embeddings, and also investigate the effect of different model architectures and languages on attention behavior.
For example，the authors find that 1) learnable absolute position embedding contains sinusoid-like waves, 2) attention heads are able to extract periodic components from the position embedding in the hidden states, and 3) the self-attention mechanism is responsible for adjusting the phase of the periodic components in both the query and the key, thus influencing the direction of attention.
These findings provide a better understanding of the mechanisms underlying position encoding in attention, which can inform future model design and development.

**Reasons To Accept:**

1. Thorough analysis of attention mechanisms. The paper's findings on the impact of position embeddings on attention is particularly valuable.
2. Clear presentation of experimental results and findings.
3. The authors' use of theoretical interpretations and visualization techniques to illustrate attention behavior is particularly helpful in conveying their findings to readers.

**Reasons To Reject:**

1. The authors' findings are related to the models and training objectives used in their study, and may not be entirely generalizable. However, the detection and analysis methods proposed by the authors can be applied to other models.

**Reproducibility:**

3: Could reproduce the results with some difficulty. The settings of parameters are underspecified or subjectively determined; the training/evaluation data are not widely available.

**Reviewer Confidence:**

4: Quite sure. I tried to check the important points carefully. It's unlikely, though conceivable, that I missed something that should affect my ratings.

---

### Official Review · Reviewer_5Jmf · 2023-08-05

**Soundness:** 4

**Excitement:**

4: Strong: This paper deepens the understanding of some phenomenon or lowers the barriers to an existing research direction.

**Paper Topic And Main Contributions:**

This paper analyzes the mechanism of relative positional embeddings and shows that the relative positional dependence of attention emerges due to some factors. Besides, the word embedding is also a factor that enables inference based on relative position for the attention strongly concentrated on the adjacent tokens.

**Questions For The Authors:**





**Reasons To Accept:**

This paper introduces an interesting work on the relationship between self-attention and position embeddings. The factors proposed in the paper are based on several experiments and the results are persuasive.

**Reasons To Reject:**

There are some problems in this paper

1.  The relationships between different sections are not clear and authors should give a clear description of the relationship in the introduction. In the Introduction, these factors are put in Section 4. However, Section 3 also shows the factors of RPE. Besides Section 4 focuses on nearby tokens and Section 5 focuses on adjacent tokens, while adjacent tokens can be viewed as one part of nearby tokens. It is better to combine Section 4 and 5.

2. Authors should make the equation more prescriptive. Line 244, $X$a->$X_a$, $Y$b->$Y_b$. Line 329, $q$s->$q=[q_n]$, $k$s->$k=[k_n]$

**Reproducibility:**

3: Could reproduce the results with some difficulty. The settings of parameters are underspecified or subjectively determined; the training/evaluation data are not widely available.

**Reviewer Confidence:**

3: Pretty sure, but there's a chance I missed something. Although I have a good feel for this area in general, I did not carefully check the paper's details, e.g., the math, experimental design, or novelty.

---

> ### Author Rebuttal · Authors · 2023-08-27
>
> We will fix the problems mentioned in the Reasons To Reject section in camera-ready version.
>
> > The relationships between different sections are not clear and authors should give a clear description of the relationship in the introduction. In the Introduction, these factors are put in Section 4. However, Section 3 also shows the factors of RPE.
>
> In the camera-ready version, an description of the standpoint in Section 3 will be added to the Introduction.
>
> Section 3 demonstrates the subject of analysis in this paper. It is a phenomenon in which the type of attention using Absolute Position Embedding depends on the relative positions. The mechanism is then analyzed in sections 4 and 5.
>
> > Besides Section 4 focuses on nearby tokens and Section 5 focuses on adjacent tokens, while adjacent tokens can be viewed as one part of nearby tokens. It is better to combine Section 4 and 5.
>
> Adjacent tokens are indeed included in the nearby tokens, but the mechanism by which attention is focused on each may be different, as shown in §4.3 and Fig. 9.
>
> > Authors should make the equation more prescriptive. Line 244, $Xa$ -> $X_a$, $Yb$ -> $Y_b$.
>
> Since $a$ and $b$ are weight vectors that compose the synthetic variables $Xa$ (and $Yb$ ) as the product of $X$ and $a$ (and $Y$ and $b$ ), the original notation is correct.
> In the camera-ready version, we refer to $a$ and $b$ as parameters.

---

### Official Review · Reviewer_fPAJ · 2023-08-05

**Soundness:** 3

**Excitement:**

3: Ambivalent: It has merits (e.g., it reports state-of-the-art results, the idea is nice), but there are key weaknesses (e.g., it describes incremental work), and it can significantly benefit from another round of revision. However, I won't object to accepting it if my co-reviewers champion it.

**Missing References:**

- In addition to Clark+'19, Kovaleva+'19 should be mentioned as a representative attention analysis
    - Kovaleva+, Revealing the Dark Secrets of BERT (EMNLP2019)

**Paper Topic And Main Contributions:**

Positional encoding is important in Transformer architecture, and until a few years ago, learnable absolute position embedding (APE) was often used (e.g., BERT, RoBERTa, GPT-2).

Clark+'19 reported that some attention heads in BERT attend to context words according to their relative positions; Ravishankar&Søgaard+'21 reported that some columns of absolute position embeddings are periodic; Chang+'22 reported that the position information is encoded in a hidden representation while remaining periodic.
However, it is not clear how periodicity is used in models, which is what this paper attacks.

Specifically, this paper showed that several attention heads in RoBERTa realize an attention pattern that depends on relative position by extracting the periodic components derived from APE from the hidden representation with shifting the phase in query and key transformation.

**Questions For The Authors:**

- A. What is the substantial contribution to the community from these findings? For example, could they lead to any ideas to improve the Transformer architecture?
- B. Clark+'19 reported that most heads pay little attention to themselves and that there are heads that focus heavily on adjacent tokens, especially in the earlier layers. However, this paper shows that there are several heads that focus on themselves and that heads focusing heavily on adjacent token heads were found in various layers. What is the reason for this discrepancy?
- C. Nowadays, relative position embedding is often used instead of APE. Can this paper provide insights into the reasons for the superiority of relative position embedding? For example, can we interpret that relative position embedding can capture other language information richer than APE because there is no need to extract relative positions in the query or key transformation matrices?

**Reasons To Accept:**

- Analyses use convincing and solid mathematical tools.
- Mechanism for realizing relative position-dependent attention patterns from absolute position embedding is really interesting.

**Reasons To Reject:**

- Limited Experiments
  - Most of the experiments (excluding Section 4.1.1) are limited to RoBERTa-base only, and it is unclear if the results can be generalized to other models adopting learnable APEs. It is important to investigate whether the results can be generalized to differences in model size, objective function, and architecture (i.e., encoder, encoder-decoder, or decoder). In particular, it is worthwhile to include more analysis and discussion for GPT-2. For example, I would like to see the results of Figure 2 for GPT-2.
  - The input for the analysis is limited to only 100 or 200 samples from wikitext-2. It would be desirable to experiment with a larger number of samples or with datasets from various domains.
- Findings are interesting, but no statement of what the contribution is and how practical impact on the community or practical use. (Question A).
- Results contradicting those reported in existing studies (Clark+'19) are observed but not discussed (Question B).
- I do not really agree with the argument in Section 5 that word embedding contributes to relative position-dependent attention patterns. The target head is in layer 8, and the changes caused by large deviations from the input, such as only position embedding, are quite large at layer 8. It is likely that the behavior is not such that it can be discussed to explain the behavior under normal conditions. Word embeddings may be the only prerequisites for the model to work properly rather than playing an important role in certain attention patterns.
- Introduction says to analyze "why attention depends on relative position," but I cannot find content that adequately answers this question.
- There is no connection or discussion of relative position embedding, which is typically employed in recent Transformer models in place of learnable APE (Question C).

**Reproducibility:**

3: Could reproduce the results with some difficulty. The settings of parameters are underspecified or subjectively determined; the training/evaluation data are not widely available.

**Reviewer Confidence:**

3: Pretty sure, but there's a chance I missed something. Although I have a good feel for this area in general, I did not carefully check the paper's details, e.g., the math, experimental design, or novelty.

**Typos Grammar Style And Presentation Improvements:**

- Caption of Figure 1: Attention weights of the first 15 tokens -> Attention weights for the first 15 tokens of an input text? If it is a special input text, please explain explicitly.
- ll.223-227: I cannot follow what you meant. Please provide a clearer explanation.
- ll.263-265: Should mention the agreement with Lin+'19 results here
- (minor) l.200: Consider changing the variable name PE, since PE appears to be the product of two matrices P and E.

---

> ### Author Rebuttal · Authors · 2023-08-27
>
> We first answer to question B and C and then answer to question A.
>
> > B. Clark+'19 reported that most heads pay little attention to themselves and that there are heads that focus heavily on adjacent tokens, especially in the earlier layers. However, this paper shows that there are several heads that focus on themselves and that heads focusing heavily on adjacent token heads were found in various layers. What is the reason for this discrepancy?
>
> Clark+'19 used bert-base-uncased, while we used the roberta-base. We created a bert-base-uncased version of Figure 2. The heads in which 90% of the attention matrices was assigned to the same cluster were as follows.
>
> | cluster       | heads (notation is `<layer>-<head_number>` as in Clark+'19)    |
> | :------------ | ---------------------------------------------------------------- |
> | next-to-left  | 2-5, 4-6, 7-12, 8-5                                              |
> | leftward      | 1-3, 1-4, 3-2, 4-12, 5-8, 5-11, 6-11, 7-4, 7-5, 7-10, 8-10, 8-12 |
> | next-to-right | 1-11, 2-2, 3-1, 3-10, 4-10, 5-12, 6-10, 7-7, 8-3                 |
> | rightward     | 2-7, 3-12, 4-4, 5-6, 6-1, 7-6, 8-2, 8-9, 8-11, 9-6, 9-10         |
> | itself        | 3-7, 10-7, 11-10, 11-11, 12-9                                    |
>
> For comparison, we quote the relevant part of Clark+'19: "In particular four attention heads (in layers 2, 4, 7, and 8) on average put >50% of their attention on the previous token and five attention heads (in layers 1, 2, 2, 3, and 6) put >50% of their attention on the next token".
>
> Next-to-left seems to be in agreement. However, there is still a discrepancy in that our method finds more heads in which each token focuses attention on itself. A possible reason is that **the t-offset trace (Eq. 7) emphasizes attention patterns that depend on relative positions**.
>
> For example, consider the attention matrix $A$ of size 10x10 shown below.
> This can be regarded as a case of mixed attention to relative positions and attention to specific tokens (e.g., special tokens or period).
>
> $$
> A = \begin{bmatrix}
> 0.1 & 0 & 0 & 0 & 0 & 0.9 & 0 & 0 & 0 & 0 \\\\
> 0 & 0.1 & 0 & 0 & 0 & 0.9 & 0 & 0 & 0 & 0 \\\\
> 0 & 0 & 0.1 & 0 & 0 & 0.9 & 0 & 0 & 0 & 0 \\\\
> 0 & 0 & 0 & 0.1 & 0 & 0.9 & 0 & 0 & 0 & 0 \\\\
> 0 & 0 & 0 & 0 & 0.1 & 0.9 & 0 & 0 & 0 & 0 \\\\
> 0 & 0 & 0 & 0 & 0 & 1 & 0 & 0 & 0 & 0 \\\\
> 0 & 0 & 0 & 0 & 0 & 0.9 & 0.1 & 0 & 0 & 0 \\\\
> 0 & 0 & 0 & 0 & 0 & 0.9 & 0 & 0.1 & 0 & 0 \\\\
> 0 & 0 & 0 & 0 & 0 & 0.9 & 0 & 0 & 0.1 & 0 \\\\
> 0 & 0 & 0 & 0 & 0 & 0.9 & 0 & 0 & 0 & 0.1 \\\\
> \end{bmatrix}
> $$
>
> The t-offset trace of this matrix takes its maximum value at $t=0$.
>
> As the number of input tokens is increased, the attention that depends on the relative positions becomes more emphatic.
> Thus, even if the average attention to itself is small, the t-offset trace can be maximized at relative position $t=0$.
>
> This effect is true for any relative position, not just $t=0$.
> That is, our method can detect the heads for which the actual value of attention is small, but for which attention consistently depends on a certain relative position.
>
> > C. Nowadays, relative position embedding is often used instead of APE. Can this paper provide insights into the reasons for the superiority of relative position embedding? For example, can we interpret that relative position embedding can capture other language information richer than APE because there is no need to extract relative positions in the query or key transformation matrices?
>
> In our paper, we showed that inference with APE also depended on relative positions. In other words, the goal of APE and RPE is the same: to acquire translation invariance. However, APE requires the entire model to acquire translation invariance, whereas RPE directly models the property. Thus, APEs are at a disadvantage if the inference requires translation invariance.
>
> Specifically, in order to shift the position where attentions are focused by a specific number of tokens, the APE and parameters must learn the frequency and the rotation angle, respectively, to follow Eq. 27 (Fig. 10).
>
> > A. What is the substantial contribution to the community from these findings? For example, could they lead to any ideas to improve the Transformer architecture?
>
> This could be one of the reasons why APE is not a good choice for position embedding.
>
> Assuming that the acquisition of translation invariance of the model is essential for language processing, APEs are structurally disadvantaged in acquiring this property, as noted in the response to Question C.
>
> ---
>
> > Findings are interesting, but no statement of what the contribution is and how practical impact on the community or practical use. (Question A).
>
> I completely agree that it is valuable to apply the accumulated technology to the real world. However, we believe that the goal of NLP also includes the clarification of the mechanisms by which machines process language, i.e., "Interpretability, Interactivity and Analysis of Models for NLP," the name of this track. If practicality is a requirement for acceptance, we fear that research other than proposals for new models, architectures, etc. will be at a disadvantage unreasonably.
>
> > Most of the experiments (excluding Section 4.1.1) are limited to RoBERTa-base only, and it is unclear if the results can be generalized to other models adopting learnable APEs. It is important to investigate whether the results can be generalized to differences in model size, objective function, and architecture (i.e., encoder, encoder-decoder, or decoder). In particular, it is worthwhile to include more analysis and discussion for GPT-2. For example, I would like to see the results of Figure 2 for GPT-2.
>
> This paper targets the phenomenon that, **despite that self-attention without causal mask is symmetric with respect to position, its inference is dependent on position**. It is interesting to see how decoders, which pay attention only to the preceding tokens, handle position, but this is a topic out of the scope of this paper, since the principles of causal modeling and bidirectional modeling are completely distinct.
>
> > Introduction says to analyze "why attention depends on relative position," but I cannot find content that adequately answers this question.
>
> We will correct "how and why" in L062 to "how".

---

### Meta-Review · Area_Chair_jitg · 2023-09-17

**Recommendation:** 4

**Metareview:**

The paper tries to understand how Transformer models with learned positional encodings manage to understand relative position (as seen, e.g., by relative position attention heads). The authors use DFT and PCA to show that positional embeddings encode sinusoid-like waves, and CCA along with spectral analysis for query and key in attention to show that attention to previous/next token is possible because the phases of these waves are shifted by the same number tokens. The paper’s experiments are only focused on RoBERTa models and the results are applicable only to MLM training objectives.

Overall, reviewers agreed that the discovered mechanisms are interesting and the used analysis tools look solid. The main concerns are that the experiments are limited to a single model, a very small subset of examples used in the analysis (100-200 samples from wikitext-2), very weak discussion of connections to other work. I agree with these concerns and they do not seem to be addressed in the rebuttal.

---

### Decision · Program_Chairs · 2023-10-07

**Decision:**

Accept-Main

**Comment:**

The paper tries to understand how Transformer models with learned positional encodings manage to understand relative position (as seen, e.g., by relative position attention heads). The authors use DFT and PCA to show that positional embeddings encode sinusoid-like waves, and CCA along with spectral analysis for query and key in attention to show that attention to previous/next token is possible because the phases of these waves are shifted by the same number tokens. The paper’s experiments are only focused on RoBERTa models and the results are applicable only to MLM training objectives.

Overall, reviewers agreed that the discovered mechanisms are interesting and the used analysis tools look solid. The main concerns are that the experiments are limited to a single model, a very small subset of examples used in the analysis (100-200 samples from wikitext-2), very weak discussion of connections to other work. I agree with these concerns and they do not seem to be addressed in the rebuttal.